# Chaperone-tip adhesin complex is vital for synergistic activation of CFA/I fimbriae biogenesis

Li-hui He[1], Hao Wang[2,3], Yang Liu[2,3], Mei Kang[4], Tao Li[1], Chang-cheng Li[1], Ai-ping Tong[1], Yi-bo Zhu[1], Ying-jie Song[1], Stephen J. Savarino[3,5¤], Michael G. Prouty[3]*, Di Xia[6]*, Rui Bao[1]*

**1** Center of Infectious Diseases, State Key Laboratory of Biotherapy, West China Hospital, Sichuan University, Chengdu, China, **2** Henry M. Jackson Foundation for the Advancement of Military Medicine, Bethesda, MD, United States of America, **3** Enteric Diseases Department, Infectious Diseases Directorate, Naval Medical Research Center, Silver Spring, MD, United States of America, **4** Department of Laboratory medicine, West China Hospital, Sichuan University, Chengdu, China, **5** Department of Pediatrics, Uniformed Services University of the Health Sciences, Bethesda, MD, United States of America, **6** Laboratory of Cell Biology, Center for Cancer Research, National Cancer Institute, National Institutes of Health, Bethesda, MD, United States of America

¤ Current address: Sanofi Pasteur, Swiftwater, Pennsylvania, United States of America
* Michael.g.proty2.mil@mail.mil (MGP); xiad@mail.nih.gov (DX); baorui@scu.edu.cn (RB)

**Data Availability Statement:** All relevant data are within the manuscript and its Supporting Information files.

**Funding:** This work was supported in part by the National Key Research and Development Plan (Grant 2016YFA0502700), by National Natural

## Abstract

Colonization factor CFA/I defines the major adhesive fimbriae of enterotoxigenic *Escherichia coli* and mediates bacterial attachment to host intestinal epithelial cells. The CFA/I fimbria consists of a tip-localized minor adhesive subunit, CfaE, and thousands of copies of the major subunit CfaB polymerized into an ordered helical rod. Biosynthesis of CFA/I fimbriae requires the assistance of the periplasmic chaperone CfaA and outer membrane usher CfaC. Although the CfaE subunit is proposed to initiate the assembly of CFA/I fimbriae, how it performs this function remains elusive. Here, we report the establishment of an *in vitro* assay for CFA/I fimbria assembly and show that stabilized CfaA-CfaB and CfaA-CfaE binary complexes together with CfaC are sufficient to drive fimbria formation. The presence of both CfaA-CfaE and CfaC accelerates fimbria formation, while the absence of either component leads to linearized CfaB polymers *in vitro*. We further report the crystal structure of the stabilized CfaA-CfaE complex, revealing features unique for biogenesis of Class 5 fimbriae.

## Author summary

Colonization factor antigen I (CFA/I) is a representative member of alternative chaperone-usher fimbriae in enterotoxigenic *Escherichia coli*, a major cause of travelers' diarrhea. During assembly, the tip-adhesive subunit CfaE might conceivably serve as an initiator, but the mechanism is not well understood. We demonstrate that the chaperone-tip adhesin complex CfaA-CfaE is essential in early fimbriae growth. The crystal structure of this complex reveals a specific chaperone-anchoring motif and a functional inter-domain loop in CfaE. Our findings suggest that the interaction of the CfaA-CfaE complex with CfaC is

Science Foundation of China (Grant No. 81501787, 81670008 and 81871615), by Ministry of Science and Technology of the People's Republic of China (No.2018ZX09201018-005), and National Mega-project for Innovative Drugs (2019ZX09721001-001-001). It is partially supported by the United States Army Infectious Disease Research Program Work Unit A1207 and the Henry M. Jackson Foundation for the Advancement of Military Medicine (to M.G.P.). This research was also supported in part by the Intramural Research Program of the NIH, National Cancer Institute, Center for Cancer Research. The funders had no role in study design, data collection and analysis, decision to publish, or preparation of the manuscript.

**Competing interests:** The authors have declared that no competing interests exist.

responsible for control of the usher plug domain movement and facilitates the binding of CfaE adhesin domain into the CfaC lumen. Collectively, our data demonstrate the crucial role of CfaA-CfaE in CFA/I fimbriae production, providing guidance for further development of novel agents targeting CFA/I fimbriae and against ETEC infection.

## Introduction

Human-specific enterotoxigenic *Escherichia coli* (ETEC) is a major cause of debilitating diarrhea in children of resource-limited countries and travelers to those regions. ETEC infection is initiated by the attachment of bacteria to the intestinal epithelia, which is mediated by bacterial surface adhesive fimbriae or pili[1]. Among different types of adhesive fimbriae expressed on the surface of ETEC bacteria, the colonization factor antigen I or CFA/I is most prevalent in isolated ETEC field strains and is archetypal of the Class 5 family fimbriae[2, 3]. Class 5 fimbriae are among the simplest bacterial organelles, consisting of only two types of subunits. For the CFA/I fimbriae, one subunit is the single-copy, tip-locating adhesin CfaE and the other is the major pilin CfaB, polymerizing to form the helical shaft of a CFA/I fimbria[4–6]. The helical form gives CFA/I fimbriae a spring-like property, being able to unwind and re-wind[7, 8]. In addition, the tip CfaE adhesin displays shear force-enhanced receptor binding, which is similar to the catch bond used by FimH in Type-1 pili[7, 9, 10]. These properties allow persistent attachment to the intestinal epithelia.

In addition to the two structural subunits, biogenesis of CFA/I fimbriae requires the functions of two factors, the periplasmic chaperone CfaA and the outer membrane usher protein CfaC, which bear functional similarities to components required for fimbriae that are assembled by "the chaperone-usher" (CU) pathway[11–19]. In the assembly process following a classic CU pathway, the chaperone captures an incompletely folded pilus subunit in the periplasm via the donor-strand complementation (DSC) mechanism, in which a β strand (G1 donor-strand) of the chaperone fits into the hydrophobic groove of a subunit, featuring 5 interaction sites (P1-P5 sites). This groove is the result of incomplete folding of the immunoglobulin (Ig)-like subunit due to the missing strand[20]. The DSC is facilitated by interactions between the subunit carboxyl terminus and a pair of basic residues from the chaperone[17, 18]. During assembly, the chaperone G1 strand complementing a subunit is replaced by the N-terminal extension (Nte, also called Gd strand) of the following subunit. This process, termed donor-strand exchange (DSE), takes place at the site of the outer-membrane usher[17, 18, 21].

Distinct from fimbriae assembled by the typical CU pathway, the major subunit CfaB of CFA/I fimbriae features a longer hydrophobic groove with an additional P0 site available for interaction with the chaperone. The chaperone CfaA also has a unique set of residues (N180 and Y182) in the cleft between its two lobes for CfaB C-terminal anchoring[22, 23]. Therefore, the CfaA chaperone has been grouped into a separate periplasmic FGA chaperone subfamily (chaperone F1-G1 loop Alternate) to distinguish it from the previously designated FGL (chaperone F1-G1 loop Long) and FGS (chaperone F1-G1 loop Short) subfamilies[23].

The tip-localized adhesins are required for the initiation of assembly for various fimbriae[24–29], including the Class 5 fimbria assembly[30]. Presumably, it is the CfaA-CfaE heterodimer that initiates CFA/I fimbria assembly, because CfaE is unstable when expressed alone[31]. How the CfaA-CfaE dimer is recognized by the usher protein CfaC remains unclear, but the formation of the CfaE-CfaC complex is assumed to be required for the subsequent incorporation of thousands copies of the major pilin subunit CfaB to form a helical rod of CFA/I fimbriae.

Previous studies on CfaE have focused mainly on its structure, host cell-interaction and the mechanism of persistent binding[29, 32, 33]. The role played by CfaE in CFA/I fimbria

assembly remains unclear. In this study, we sought to reconstitute CFA/I fimbriae *in vitro* to enable investigation of the assembly process. Toward that goal, we engineered a mutant CfaA variant to stabilize the CfaA-CfaE and CfaA-CfaB complexes, which allowed *in vitro* assembly of CFA/I fimbriae in the presence of isolated usher CfaC. The stabilized CfaA-CfaE heterodimer also permitted structure determination of the CfaA-CfaE complex and the structure allowed identification of a distinct chaperone-anchoring motif and a functional inter-domain loop in CfaE. Our work shows that CfaE is required for the efficient assembly of CFA/I fimbriae.

## Results

### *In vitro* assay of CFA/I assembly suggests essential functions for both chaperone-adhesin complex and usher

Like most known mono-adhesin CU fimbriae, the CfaE adhesin is composed of an adhesin domain and a pilin domain [29, 33, 34]. The N-terminal adhesin domain is responsible for host-cell receptor binding and the C-terminal pilin domain interacts with CfaA or CfaB. Thus, the pilin domain requires a donor strand from either chaperone CfaA or CfaB subunit to stabilize its incomplete Ig-like fold[17, 32]. We attempted to generate a binary complex of native CfaA and CfaE, but it was not stable in solution (S1 Fig), similar to what was observed for the native CfaA and CfaB complex[22]. Previous studies have shown that substitutions of residues in the G1 donor strand of the chaperone with small hydrophobic residues enhance stability of the chaperone-subunit heterodimer[22, 23, 35]. However, for CfaA, the single mutation T112I resulted in delayed fimbriation, a phenotype that can be rescued by the triple mutation T112I/L114I/V116I. The latter displays a similar behavior in fimbriation as the wild-type CfaA[22]. When compared to native CfaA, the mutant variant containing the triple residue substitution T112I/L114I/V116I on G1 strand (hitherto referred to as mtCfaA) formed significantly stabilized complexes with CfaE (Fig 1A), CfaB and CfaBntd (the CfaB variant lacking the N-terminal donor strand). These stabilized complexes were isolated and used in subsequent *in vitro* assays.

To reconstitute the CFA/I fimbriae *in vitro*, we incubated stabilized mtCfaA-CfaB alone (5 μM) at 25˚C or in combination with the purified recombinant CfaC (S2 Fig) and/or the stabilized mtCfaA-CfaE at a concentration of 0.25 μM each. The assay mixtures were sampled at various incubation time points and polymerization was monitored by the disappearance of the input mtCfaA-CfaB binary complex using acidic-native PAGE (AN-PAGE, Fig 1B to 1C). Samples taken after 24h incubation were further analyzed using electron microscopy (EM) with negative staining (Fig 1D to 1E). Similar to the reconstitution experiment of Type 1 fimbriae[30], CfaB self-assembly was very slow in the absence of mtCfaA-CfaE or CfaC. No regular rod-like fimbriae were observed under negative-stain EM except for some short fragments that are probably the result of self-aggregation of mtCfaA-CfaB (S3 Fig). By contrast, only in the presence of both CfaC and mtCfaA-CfaE, did CfaB polymerization progress efficiently, forming typical fimbria rods. Without the tip-adhesin subunit, CfaC was able to prevent CfaB self-assembly into off-pathway trimer and to extend CfaB polymer, although at a much lower efficiency. A similar observation was made when mtCfaA-CfaB was incubated with mtCfaA-CfaE only. These results indicate that CfaB polymerization can be initiated by either usher or chaperone-adhesin complex, but correct and efficient fimbria assembly requires the synergy of both components.

### Crystal structure of <sup>mt</sup>CfaA in complex with CfaE

With the functional, stabilized mtCfaA-CfaE purified, we proceeded to crystallize the complex as part of our effort to understand the underlying mechanism and the requirement of CfaE as

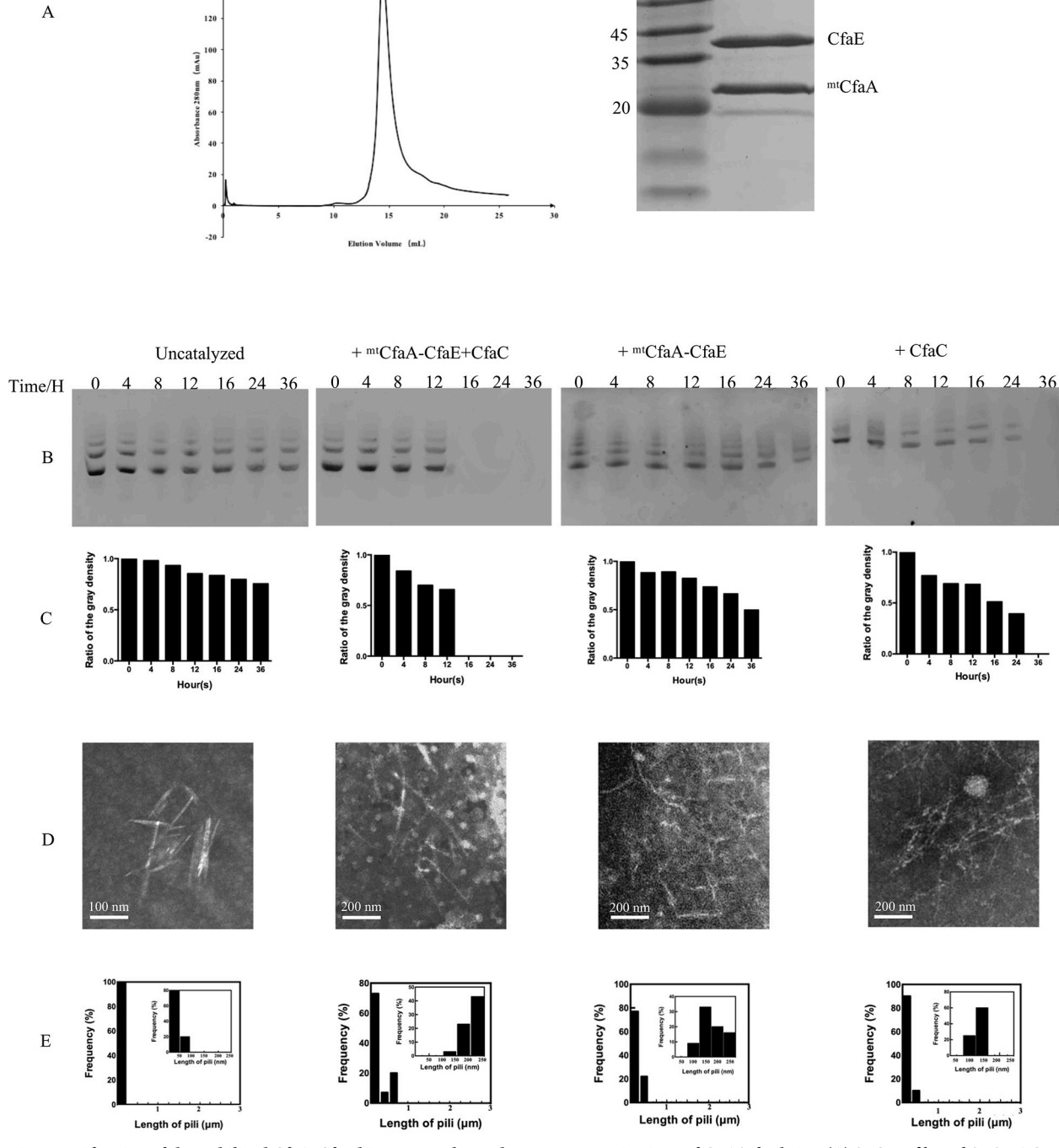

**Fig 1. Purification of the stabilized CfaA-CfaE binary complex and *in vitro* reconstitutions of CFA/I fimbriae.** (A) SEC profile and SDS-PAGE of purified stabilized CfaA-CfaE complex. The stabilized CfaA-CfaE complex was eluted as a single peak in a SEC run. SDS-PAGE of the SEC peak resolved components of the CfaA variant at 25 kDa and the CfaE at 39 kDa. (B to E) *In vitro* reaction of the assembly of CFA/I fimbriae. (B) The reaction was followed by the disappearance of the CfaA-CfaB complex on AN-PAGE, the protein ladders resulted from aggregations of CfaA-CfaB (S3 Fig). (C) The quantifications of ladders on AN-PAGE in Fig 1B. (D) Electron micrographs showing the results of the *in vitro* fimbriae assembly assay under various conditions. Samples were taken after 24 hours of incubation. Representative CfaB polymers under various conditions are indicated by arrows. (E) The length distribution histograms of generated pili. The insert panels show the frequency with pili length of 1-250nm.

an initiator of CFA/I assembly. The structure was determined by the molecular replacement method using native CfaA (PDB:4NCD) and CfaE (PDB:2HB0) as search models. There are two identical copies of the mtCfaA-CfaE complex in one crystallographic asymmetric unit. The structure was refined to 2.77Å resolution with $R_{work}$ of 22.5% and $R_{free}$ of 26.0% (Table 1). In the structural model of the complex, the loops containing residues 100–109 in mtCfaA and residues 311–320 in CfaE were not built because of poor electron density in those regions.

CfaA interacts with CfaE in the canonical DSC manner via a pair of parallel β-strands: one from the CfaA G1 strand and the other from the CfaE F2 strand (Fig 2A). Although the N- (residues 20–126) and C- (residues 127–220) terminal domains of CfaA remain rigid during its coupling to different subunits (the overall RMSD is 0.781Å), inter-domain movement is detected by overlaying the CfaA N-terminal domains of the CfaB and CfaE complexes as well as the apo form (Fig 2B). Compared to the mtCfaA-CfaBntd structure, the mtCfaA-CfaE structure reveals a slightly more open cleft between the two domains of CfaA to accommodate the larger CfaE adhesin (Fig 2C).

## Structural comparison between ᵐᵗCfaA-CfaE and ᵐᵗCfaA-CfaBⁿᵗᵈ

Expression of binary complexes of native CfaA-CfaB and CfaA-CfaE shows that both complexes are unstable[22]. Biochemically, the CfaA-CfaE heterodimer appeared to have a shorter

**Table 1. Statistics on qualities of diffraction data set, phasing and refined atomic model.**

| Data Collection | |
|---|---|
| Wavelength (Å) | 0.9778 |
| Space Group | R 32 |
| Unit Cell parameters $a, b, c$ (Å) $\alpha, \beta, \gamma$ (°) | $a = 217.2, b = 217.2, c = 177.1$ $\alpha = \beta = 90, \gamma = 120$ |
| Resolution (Å) | 29.52–2.774 (2.873–2.774) |
| No. Unique reflections | 40388 |
| Completeness (%) | 99.53 (98) |
| Redundancy | 7.5 (5.1) |
| Mean I/sigma (I) | 12.29 (2.46) |
| Refinement statistics | |
| $R_{work}$ | 0.2248 (0.3197) |
| $R_{free}$ | 0.2597 (0.3722) |
| No. protein atom (no hydrogen) | 8734 |
| No. no-protein atoms | 1072 |
| Average B-factor (Å2) | 52.3 |
| Rmsd for bond lengths (Å) | 0.033 |
| Rmsd for bond angles (°) | 1.6 |
| Ramachandran plot | |
| Favored (%) | 96.58 |
| Allowed (%) Dis-allowed (%) | 3.13 0.28 |

a. Numbers in parentheses are statistics of the outer shell.

b. $R_{work} = \Sigma|Fo\text{-}Fc|\Sigma Fo$ where $Fo$ and $Fc$ are the observed and calculated structure factors respectively.

c. $R_{free}$ is calculated for a test set of reflections randomly excluded from refinement.

d. B-factors are given with contribution from TLS tensors included. Rmsd stereochemistry is the deviation from ideal values. Rmsd B-factors is deviation between bonded atoms.

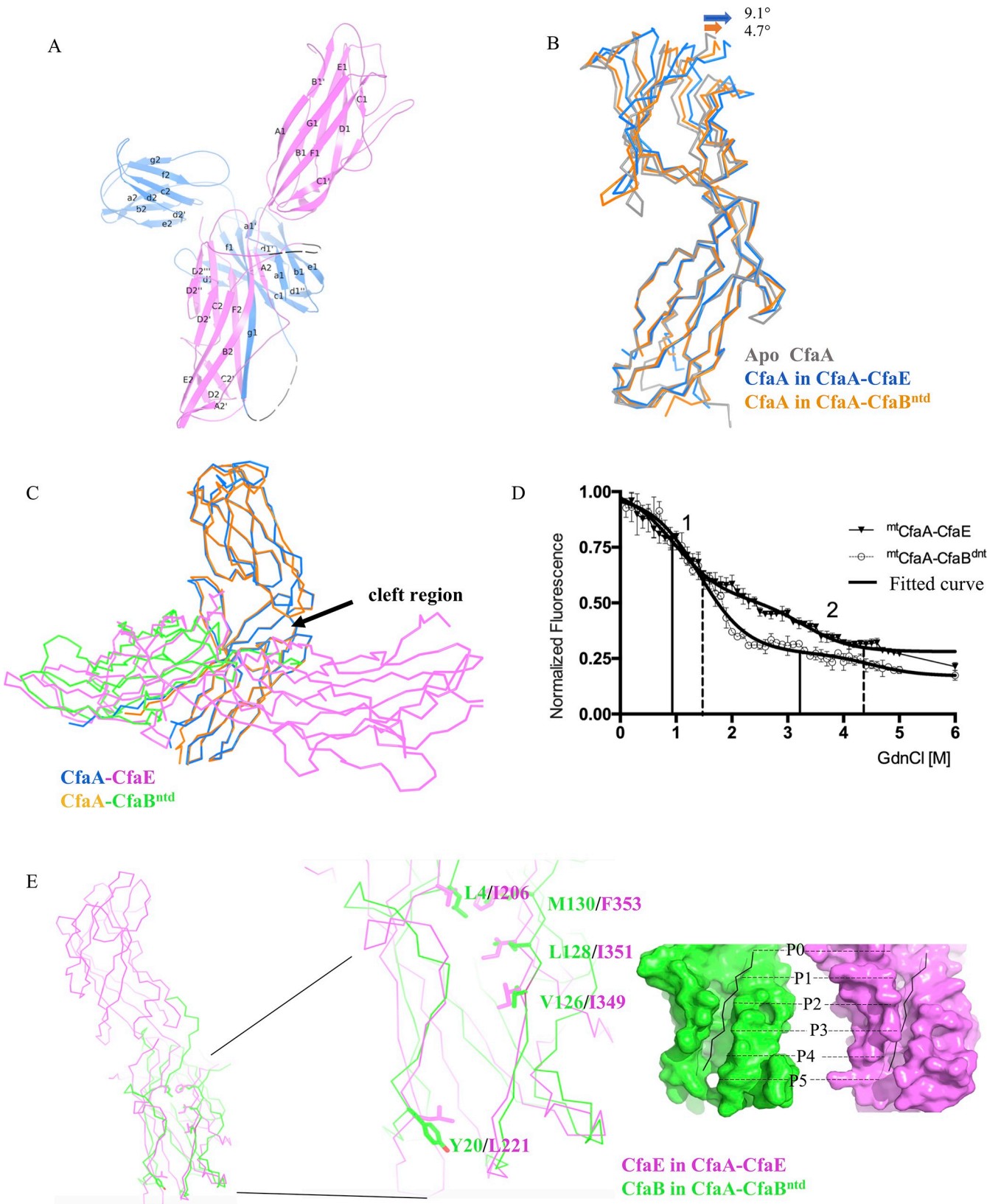

**Fig 2. Crystal structure of the CfaA-CfaE binary complex and comparison with CfaA-CfaB<sup>ntd</sup> (PDB:4Y2O).** (A) Overall structure of the mtCfaA-CfaE binary complex in cartoon representation. The CfaE is colored in magenta and the CfaA is in blue. The disordered region in CfaA (residues 100–109) and CfaE (residues 311–320) are drawn as black dashed lines. The secondary structures of each protein are labelled following the convention of previous publications. The diagram was produced with PyMOL [69] (http://www.pymol.org). (B) Superposition of the structures of CfaA from mtCfaA-CfaBntd and mtCfaA-CfaE to the apo CfaA (PDB:4NCD) based on overlying N-terminal domains of CfaA. The apo CfaA is in light grey, the CfaA from mtCfaA-CfaE is in blue and that from mtCfaA-CfaBntd is in orange. The direction of the angular movement of the C-terminal domains of CfaA upon binding of CfaB and CfaE are indicated. (C) A different view of the superposition in (B) showing the angular movement of the two domains of CfaA opening up by binding to CfaB or CfaE. (D) Recording of changes in aromatic residue fluorescence at 320 nm as a function of concentration of GdnCl for mtCfaA-CfaBntd (white circular) and mtCfaA-CfaE (black triangle). The numbers 1 and 2 indicate the 2 inflection points in mtCfaA-CfaBntd (solid line) and mtCfaA-CfaE (dash line), respectively, and the thick lines are the fitted curves generated by GraphPad software Prism 7 (https://www.graphpad.com/). (E) Superposition of the three structures in (B) also brings the pilin domain of CfaE and CfaB into alignment, allowing structural comparison of the pilin domain of CfaE (magenta) from CfaA-CfaE with CfaB (green) of CfaA-CfaBntd. The G1-binding hydrophobic grooves of the two subunits are shown as molecular surfaces with filled G1 peptides in black, with labelled binding pockets. The critical residues forming the hydrophobic groove are shown as stick models.

life-span compared to that of the CfaA-CfaB complex in solution, because we failed to isolate any native CfaA-CfaE. To further verify the relative stability of the two complexes, we carried out a guanidine hydrochloride (GdnCl)-induced dissociation and denaturation experiment to compare the thermodynamic stability of the stabilized mtCfaA-CfaE and mtCfaA-CfaBntd complexes, where mtCfaA refers to the CfaA variant with triple residue substitution in the G1 strand. The experiment was performed by monitoring fluorescence changes in response to the exposure of buried aromatic residues at the interface between CfaA and CfaE (W207 and F353 of CfaE, Y92 and Y120 of CfaA) and that between CfaA and CfaB (Y92, Y120 of CfaA), as the concentration of GdnCl increases. The changes in fluorescence as a function of GdnCl concentration (Fig 2D) for both complexes illustrate two cooperative transitions, representing a dissociation-coupled unfolding process[36]. The calculated mid-point dissociation concentration of GdnCl for mtCfaA-CfaE (0.966 M) is lower than that for the mtCfaA-CfaBntd (1.415 M), supporting our observation that mtCfaA-CfaE is indeed thermodynamically less stable than mtCfaA-CfaBntd.

To further explore the difference in thermodynamic stability of these complexes, we next sought to compare the structures of the complexes. Structure-based sequence alignment of CfaB and pilin domain of CfaE shows a very low sequence identity of 13%, suggesting differences in coupling to CfaA between the CfaE and CfaB. The structure of the CfaA-CfaE allowed us to compare the interactions between adhesin and chaperone with those between pilin and chaperone, which identified two major interaction sites on the subunits for chaperone: the DSC grooves and the C-terminal anchorage.

The structures of CfaA-CfaE and CfaA-CfaB show that CfaE and CfaB accommodate the G1 donor-strand of CfaA in different ways (Fig 2E). For the P0, P1 and P5 sites of the DSC groove, residues L4, V126, L128, M130, Y20 in CfaB are replaced by the smaller I206, I349, I351, F353, L221 residues in CfaE. The differences in hydrophobic side chains of these residues in CfaE, especially for the residues F353 (M130 in CfaB) at P0 and Y20 (L221 in CfaB) at P5, lead to a shallower, narrower groove for binding to the CfaA G1 strand, which is consistent with weaker interactions between the adhesin and chaperone.

The second site of contact between the chaperone and subunit is in the cleft of the chaperone to accommodate the C-terminal residues of the subunit. This site has long been recognized to play an important role in fimbria assembly[17, 18, 37]. Structure-based sequence alignment of CfaB and the pilin domain of CfaE shows a C-terminal extension of three residues (358-QTL-360) for CfaE. In other words, there are four residues in CfaE likely to engage in CfaA interaction (Fig 3A). The main-chain and side-chain oxygen atoms of CfaE S357 form hydrogen bonds with the essential cleft residue pair N180'/Y182' and the conserved residue E46' (residues from CfaA are marked with'), while Q358, T359, and L360 interact with T43', N180', and N203' through main-chain and side-chain hydrogen bonds. This C-terminal

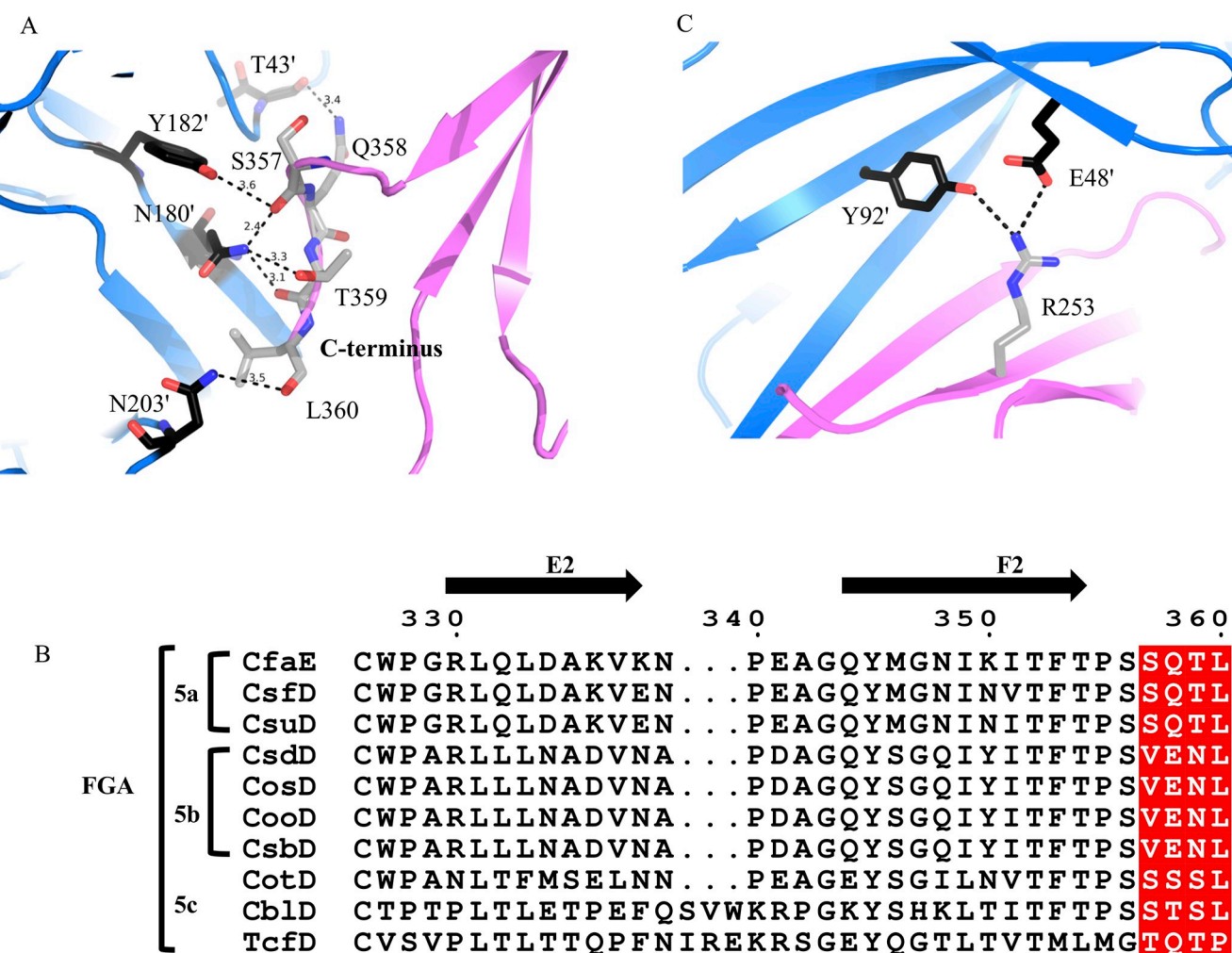

**Fig 3. Unique features of ᵐᵗCfaA-CfaE binary complex.** (A) The last four residues (magenta) of the C-terminus of CfaE (S357, Q358, T359, L360) interact with residues (blue) in the cleft region (T43', N180', Y182' and N203') of CfaA. The residues are shown as stick models. (B) Sequence alignment of C-terminal segments from different adhesin subunits of the Class 5 fimbriae. The four C-terminal residues unique to the class are shaded red. Sequences were aligned using the program ClustalX [70], and the alignment was made using the online ESPript 3.0 server [71] (http://espript.ibcp.fr/ESPript/ESPript/). (C) Interactions of R253 (magenta) in CfaE with CfaA (blue).

extension is present in all the adhesins of FGA fimbriae (Fig 3B), indicating a possible role of this sequence motif in the function or assembly of this fimbria family. Indeed, substitutions of any one of the last four C-terminal residues to alanine results in a complex pattern of effects on CFA/I fimbriation (Fig 4A, Table 2). While the S357A and Q358A mutations reduce fimbriation, mutations T359A and L360A seem to have no discernable impact on fimbriation. Given the reduced binding affinity between CfaA and CfaE compared to the CfaA-CfaB complex, this C-terminal sequence motif of CfaE is more likely a regulatory element than a structural one.

Nearby the adhesin subunit C-terminus, residue R253 of the C2' strand of CfaE interacts with residues E48' and Y92' from CfaA, contributing to anchorage of the subunit to the cleft region (Fig 3C). This interaction is also observed for the CfaA-CfaB complex mediated by K48 of CfaB. These interactions involved in CfaE anchorage to the chaperone provide an additional feature differentiating the FGA family periplasmic chaperone from those of FGL and FGS fimbriae[23].

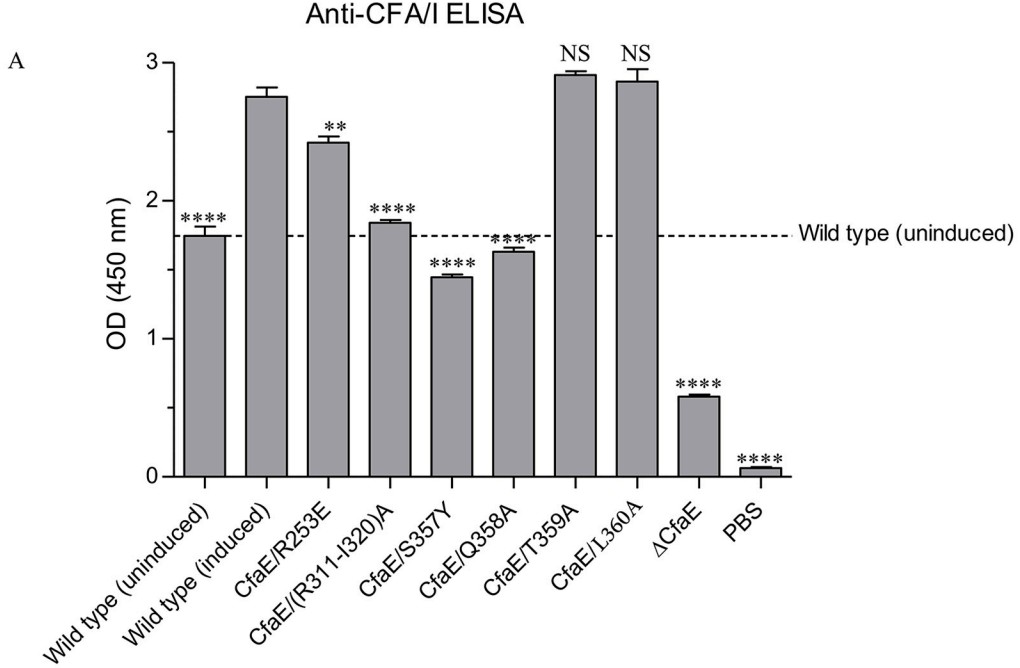

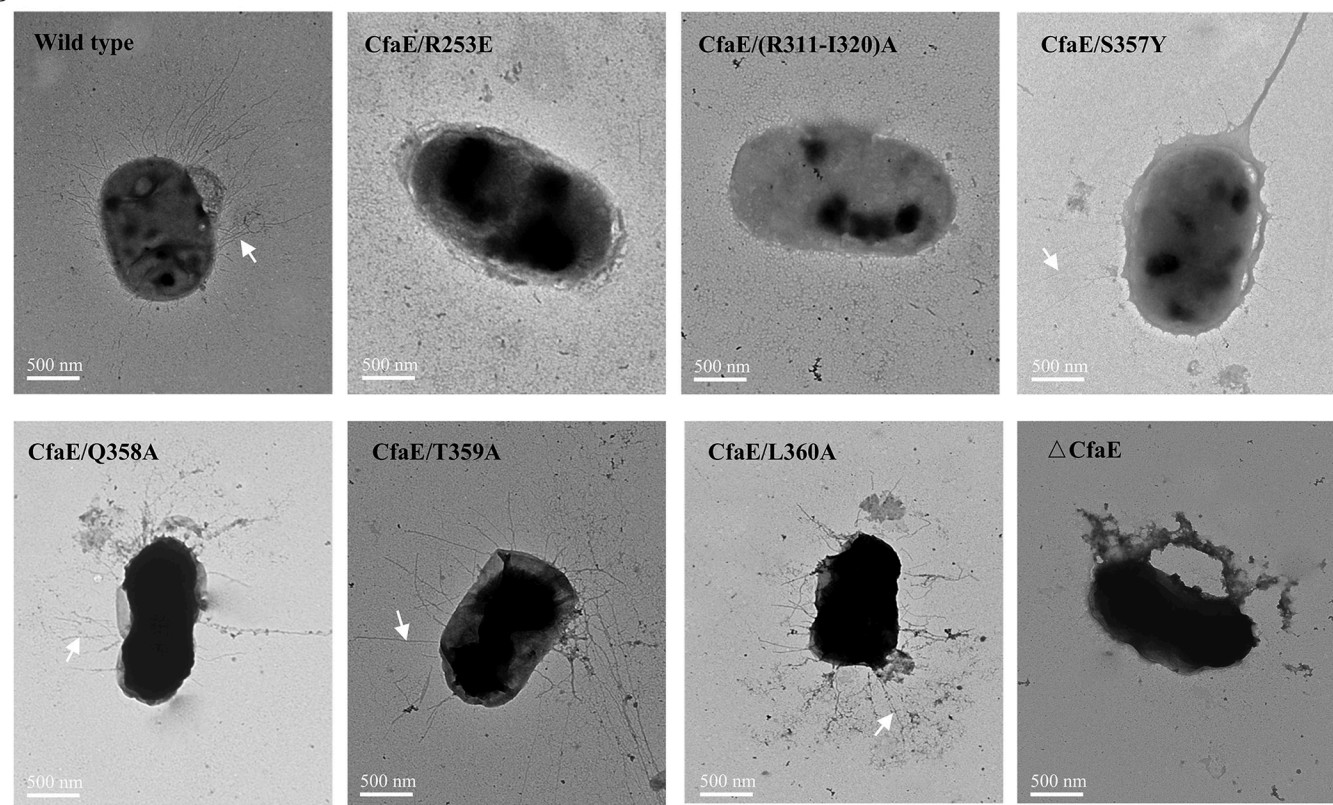

**Fig 4. Effects of CfaE mutations on CFA/I fimbriae assembly and function.** (A) The amount of CFA/I fimbriae containing CfaE variants expressed on the bacterial surface was detected by bact-ELISA using a specific antibody against subunit CfaB after the induction of CFA/I operon. (B) Negative stain EM images of CFA/I on the surface of bacteria containing CfaE mutations. The scale of each EM picture is indicated. The white arrow heads in Panel A indicate CFA/I fimbriae. The scale bar is 500 nm.

**Table 2. Effects of CfaE mutations at $^{mt}$CfaA-CfaE interface.**

| Mutations | MRHA | Location | Conservation | Possible function |
|---|---|---|---|---|
| R253E | - | C2 | Partial(5a) | Interaction with chaperone |
| (R311-I320) A | - | D2-E2 | Partial(5a,5b) | Interaction with usher |
| S357Y | - | C-terminal | Partial(5a,5c) | Anchoring on chaperone |
| Q358A | + (12) | C-terminal | Partial(5a) | Function unknown |
| T359A | + (24) | C-terminal | Partial(5a) | Function unknown |
| L360A | + (12) | C-terminal | Invariant | Function unknown |
| ΔCfaE | - | - | - | |
| Wild type (induced) | + (32) | - | - | |
| Wild type (uninduced) | - | - | - | |

a + and–signs indicate positive or negative MRHA at the highest bacterial concentration tested ($A_{600} = 40$).

b Numbers in parentheses indicate MRHA titers for the corresponding bacteria mutants. The values were the mean values performed at least in duplicate.

## Structural difference between dsc- and dse-CfaE

In previous studies of self-complemented and Gd-strand complemented CfaE (dseCfaE) structures, the adhesin and pilin domains aligned head to tail with the angles formed between the adhesive domain and the pilin domain, which we previously referred to as the joint angle defined between the longest inertial vectors of each domain. This angle is at nearly 173° for dseCfaE (Table 3), resulting in an apparent rigid cylinder[29, 32]. Upon binding to CfaA, the dscCfaE has its two domains re-arranged with the joint angle measured at 161.4° (Fig 5A). This angled arrangement of the two domains appears to be necessary to avoid physical contacts with the A1 and G1 strands of CfaA and is reminiscent of the structure of a CfaE mutant (G168D) that exhibits tighter receptor binding at low shear force[33] (Table 3). Similar interdomain rotation was also found in FimH of Type 1 fimbriae by comparing its chaperone-G1-strand-complemented and subunit-Gd-strand-bound forms[10, 18]. For Type 1 fimbriae, this angled domain arrangement not only provides an appropriate conformation for tip-adhesin to pass through the usher pore but may also preserve its folding energy for assembly initiation[38, 39]. Without energy input in the bacterial periplasm, CU fimbria systems depend on the subunit folding energy for its assembly, which is accomplished by the function of chaperones acting as subunit-folding regulators to trap the subunits in high-energy folding intermediates and to prevent self-polymerization of subunits in the periplasm[14, 40]. Thus, subsequent CfaB incorporation, membrane translocation and helical packing is driven by the energy from repeated DSE events[40, 41].

**Table 3. Interdomain interface analysis as measured by interface area, joint and twisting angles between the adhesion domain (AD: Residues 23–200) and pilin domain (PD: Residues 201–360).**

| Subunits | Interaction area | Joint angle | Twisting angle |
|---|---|---|---|
| mtCfaA-CfaE | 339.6Å2 | 161.4° | 145.0° |
| dseCfaE G168D | 444.7Å2 | 163.8° | 164.3° |
| dseCfaE | 692.1Å2 | 172.7° | 169.4° |

a. The interface area is calculated using the PDBePISA server (http://www.ebi.ac.uk/msdsrv/prot_nt/pistart.html).

b. The joint angle between the AD and PD is defined between the longest inertial vectors of each domain.

c. The twisting angle between AD and PD is based on the transformation matrix obtained from structural alignment between domains via PDBeFold server (http://www.ebi.ac.uk/msd-srv/ssm/). The angle represents a rotation in polar space around an axis to bring two domains into superposition.

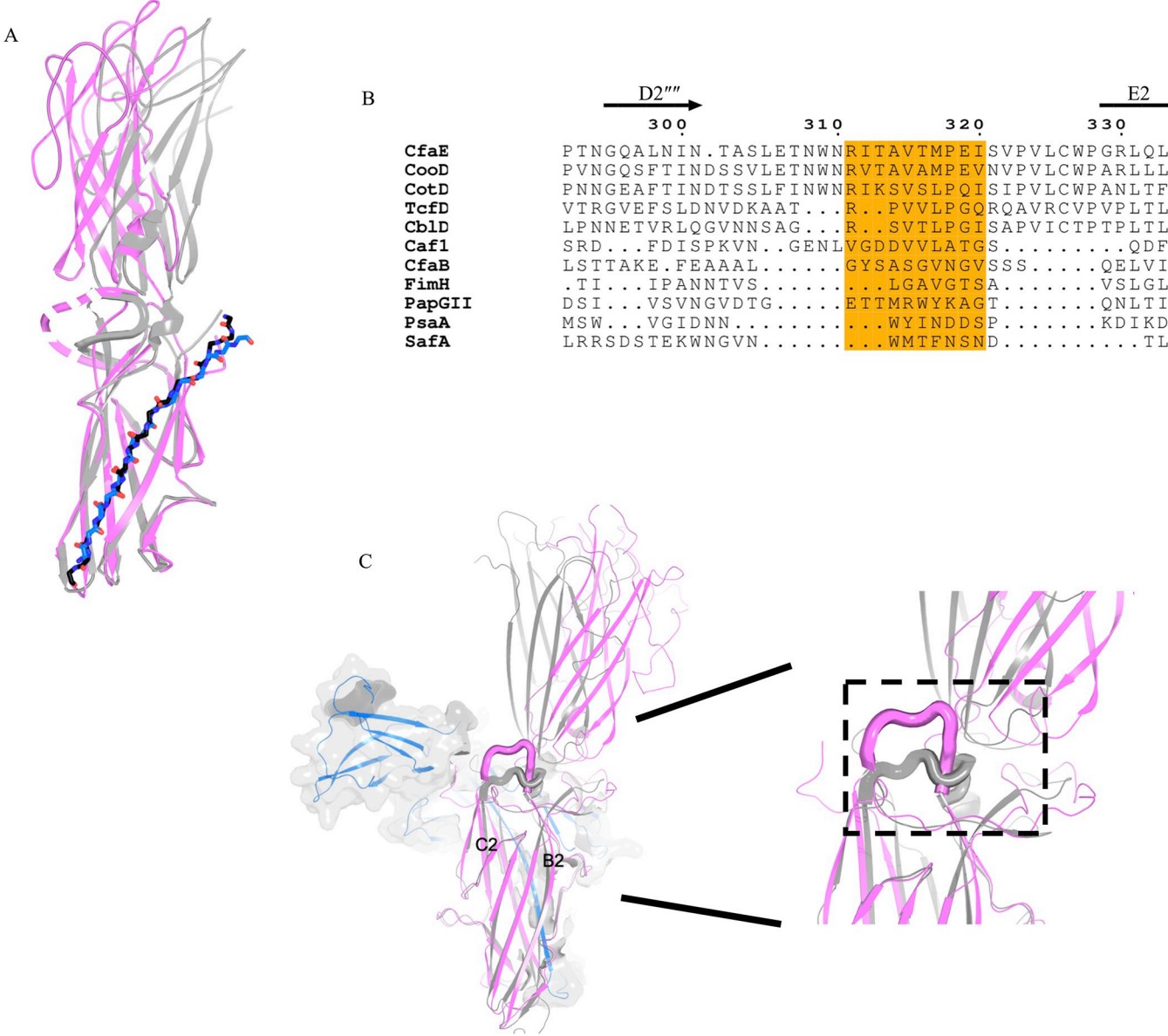

**Fig 5. Inter-domain motions of CfaE.** (A) Superimposed cartoon models of the pilin domain of CfaE from mtCfaA-CfaE (magenta) and CfaB-CfaE (gray) (PDB:3F83). The Gd strand of CfaB is in black sticks and the G1 stand of CfaA is in blue. The disordered loop 301–321 in mtCfaA-CfaE is shown as dashed line. (B) Sequence alignment of the flexible loop region, shaded in orange, from different adhesin subunits. Besides FGA fimbrial adhesins, FimH from Type I pili, PapG from P-pili, SafA from Salmonella atypical fimbriae and PsaA from the pH 6 antigen on the surface of *Yersinia pestis* are also included in the alignment. (C) Superposing the pilin domain of CfaE of the mtCfaA-CfaE complex (magenta) with that of CfaB-CfaE complex (gray), the CfaE loop (residues 241–248) connecting β-strands B2 and C2 and shown in thicker lines involved in domain interactions.

Comparing dscCfaE with dseCfaE also reveals large structural differences at the interfaces between adhesin and pilin domains. In dscCfaE, the loop (residues 304–321) connecting strands D2" and E2 is disordered, whereas in dseCfaE, this loop is ordered, wrapping around the joint and interacting with both adhesin and pilin domains (Fig 5A). Considering different lengths and compositions in corresponding regions of FGA and other fimbria adhesins[4, 9, 22, 28, 42–49] (Fig 5B), this motif may represent a unique functional site in FGA fimbriae.

Indeed, mutations introduced into this loop have a dramatic impact on the fimbria biogenesis (see below).

Another loop (residues 241–248) connecting β-strands B2 and C2 undergoes a large conformational change, with the β-strands extended and wedged between the two domains (Fig 5C), which could cause or be a consequence of the angled arrangement of these domains.

### Impact of CfaE mutations on CFA/I fimbriation

In order to validate the roles of the residues identified in the structural analysis, we investigated the impact of mutations to these residues on CFA/I fimbriation. Toward that goal, we used the vector pMAM2, which encodes the entire CFA/I operon, and generated site-directed mutations in CfaE. We evaluated the effects of these mutations on extracellular CFA/I fimbriation of the bacteria by competitive anti-CFA/I ELISA (bact-ELISA), by negative-stain EM, and by mannose-resistant hemagglutination (MRHA). Mutations were designed to specifically test the role of CfaE in fimbria biogenesis on the basis of our structural analysis, including mutating the whole CfaE (△CfaE), alteration to the chaperone anchorage and changes to the loops that displayed altered conformations in dse- and dsc-CfaE.

The △CfaE mutant had no extracellular CFA/I fimbriae by EM of negatively stained fimbria, had a negative MRHA phenotype, and no signal in bact-ELISA, validating the indispensable role of CfaE in initiating CFA/I fimbria assembly (Fig 4 and Table 2). To study the two sites for chaperone anchorage, we first mutated the C-terminal extension (SQTL) (S1 Table). When S357 was mutated to a tyrosine, the CFA/I fimbriae production was reduced to the uninduced levels, and no MRHA was observed, indicating the critical importance of a precise fit of S357 in the chaperone cleft in order to maintain the hydrogen bonding network. Mutations of the three C-terminal residues, Q358A, T359A or L360A, resulted in varying degrees of alteration in CFA/I fimbriation (Fig 4 and Table 2). The second site for CfaE anchorage unique to FGA chaperones is R253, which appears to interact strongly with the E48' and Y92' of CfaA. Indeed, a single R253E substitution in CfaE led to loss of CFA/I fimbriation by negative-stain EM and to negative MRHA, confirming its importance in anchoring CfaE to the chaperone cleft (Fig 4 and Table 2).

The structure of dscCfaE revealed interesting conformational changes compared to dseCfaE. These changes are concentrated at the interface between the adhesin and pilin domains. One loop that connects strands D2" and E2 (residues 304–321) becomes disordered in dscCfaE. We made alanine substitutions to a portion of this loop from R311 to I320 and observed no fimbriation on the cell surface by EM, which corroborated the results of bact-ELISA and MRHA (Fig 4 and Table 2). Since the loop is only flexible in dscCfaE, one possibility is that it interacts with the usher CfaC during CFA/I biogenesis.

## Discussion

### An *in vitro* system to study CFA/I fimbriae assembly

In this work, we established an *in vitro* system capable of producing native-like CFA/I fimbriae using isolated components, facilitating our investigation into the assembly process of CFA/I fimbriae. Central to this system is the engineered mutant chaperone mtCfaA that bears three mutations (T112I/L114I/V116I). This mutant variant is able to stabilize interactions with adhesin CfaE and the major pilin subunit CfaB, making possible their purification as heterodimeric complexes, structure determination of the CfaA-CfaE complex, and investigation into the assembly process.

As demonstrated previously[22, 23], mutations in the G1 strand of the CfaA chaperone often reduce the rate of CFA/I fimbria assembly. Our *in vitro* system nevertheless captures the

assembly process within a manageable time frame, allowing detection of the assembled pili and byproducts by various methods such as EM and AN-PAGE. In particular, our preliminary experiments recapitulated the important role of CfaA-CfaE binary complex as an initiator of the CFA/I assembly. Conceivably, this *in vitro* system will facilitate future investigation into the CFA/I fimbria assembly process by site-directed perturbations of each individual component.

## Roles of the CfaA chaperone in CFA/I fimbria assembly

One observation reported in studies of chaperone CfaA is the apparent unstable interactions of the chaperone with subunits CfaB and CfaE when purified. Previous study of the CfaA-CfaB complex concluded that CfaA mainly functions in preventing CfaB from going into off-pathway assembly[22]. In this work, we showed by co-expression and by GdnCl-induced dissociation and unfolding experiments that the CfaA-CfaE heterodimeric complex is even more unstable than the CfaA-CfaB complex (Fig 2D). This lower stability of the CfaA-CfaE binary complex seems consistent with the notion that CfaA traps CfaE in a higher energy state that makes dissociation easier, as manifested by the wider opening of the two lobes of CfaA (Fig 2B) and by the angled arrangement of the two domains of CfaE (Fig 5A). The fact that CfaA has a lower affinity toward CfaE, as compared to that for CfaB, seems to further suggest that (1) correct initiation of the fimbria assembly may need a suitable conformation on the part of CfaE, which is obtained by associating with CfaA, and (2) once fimbria assembly is initiated, elongation ensues very rapidly.

The CfaA-CfaE structure suggest functional roles for unique features in CfaE including the extended C-terminal motif for CfaA cleft anchoring and the interactions with the CfaA N-terminus by the flexible elbow of CfaE (310–320). While mutations in these CfaE elements did not impair binding to CfaA, they affected assembly (Table 2). Therefore, the additional interactions may help to guide the conformation of CfaE and not necessarily to increase the binding strength.

It is well known that multiple CU fimbria species are co-expressed in a single bacterial cell, each with its own cognate chaperone and usher for assembly[1]. Thus, chaperones appear to play an additional role by providing a safeguard against interference from components of other fimbria species. The structure of the CfaA-CfaE complex shows that, using a similar mechanism, the chaperone interacts with both CfaE and CfaB subunits through the extended DSC groove and a conserved serine residue (S357, corresponding to the S134 in CfaB), which are hallmark structural features distinguishing FGA fimbriae[22, 23].

## Implications for the CFA/I assembly mechanism

Unlike multi-subunit fimbriae for which the order of subunit incorporation into a growing pilus was shown to be pre-determined by specific interactions between different N-terminal extensions (Gd strands) and subunits[50, 51], the CFA/I system has only one type of Gd strand from CfaB. The initiation of fimbria assembly and subsequent pilus extension is conceivably determined by the characteristics of the CfaA-CfaE complex and its interaction with the CfaC usher. The most prominent difference between the structure of CfaA-CfaE and that of CfaA-CfaB is the adhesin domain of CfaE, which presumably is the first to bind the usher CfaC and to go across the usher pore, as seen in the structure of the initiation complex of Type 1 fimbriae[38].

Synergistic cooperation between CfaA-CfaE and CfaC appears essential for pilin subunits to polymerize into correct helical rods. This interaction may involve the R311-I320 loop of CfaE. Considering that many poly-adhesive fimbriae such as the Psa fimbriae of *Y. pestis* do not require an extra two-domain tip-subunit to initiate fimbriae assembly[42], this pilin loop

region (Fig 5B) possibly represents a recognition site and a regulatory motif for the usher. Further studies are needed to determine whether this motif is responsible for control of the usher plug domain movement.

Based on available biochemical and structural analysis of Class 5 fimbriae and studies of usher proteins in the literature[18, 34, 38, 39, 44, 52–55], we propose a scheme of synergistic chaperone-adhesin/usher-catalyzed CFA/I fimbria assembly (Fig 6). In this model, the interaction of the CfaA-CfaB complex with CfaC could not form functional CFA/I fimbriae. Instead this interaction leads to unproductive off-pathway polymerization of CfaB subunits. In addition, structural studies on Type 1 fimbria initiation complex (FimD-FimC-FimH) and elongation complex (FimD-FimC-FimF-FimG-FimH) as well as PapC C-terminal domain show that the domain binds more tightly to the usher pore in the initiation complex compared to the elongation complex[38, 39, 56]. Based on those observations, we hypothesize that the interaction of the CfaA-CfaE complex with CfaC leads to the displacement of the plug domain of CfaC and facilitates the binding of CfaE adhesin domain into the CfaC lumen. This mechanism seems to be energetically demanding and fits the energetic profile of the Type 1 fimbria assembly[39].

## Implications for overcoming the antibiotic resistance of pathogenic bacteria

The growing antibiotic resistance of gram-negative pathogenic bacteria has become a major clinical problem worldwide, compounded by the downward trend in the development of new antimicrobial drugs over the past few decades[57]. One issue is the lack of novel approaches to the problem. Our work could lead to alternative strategies for controlling bacterial infections by blocking bacterial attachment to host cells. Indeed, recent studies on uropathogenic *Escherichia coli* (UPEC) infection have shown that UPEC could be successfully controlled by inhibiting Fim fimbriae adhesin, providing a potential treatment for bacterial infection[58–61]. A similar approach could be used to curb ETEC infection by small molecule intervention and/or by vaccines against the ETEC adhesin[3]. Toward that goal, our *in vitro* assembly system and atomic-resolution structures of the CfaA-CfaE complexes can be utilized to aid the discovery of novel agents targeting CFA/I fimbriae.

## Methods and materials

### Cloning, mutagenesis and CFA/I fimbria expression

The coding sequences of CfaA (residues 20–238) with a C-terminal Strep-tag (WSHPQFEK), N-terminal hexa-histidine-tagged CfaE (residues 23–360), CfaB (residues 24–170) and C-terminal hexa-histidine-tagged CfaC were inserted into commercial expression vectors to generate pETDuet-1-CfaA(Strep), pCDFDuet-1-(His)$_6$-CfaE, pCDFDuet-1-(His)$_6$-CfaB, pET24-CfaC(His)$_6$, respectively. The plasmid pMAM2 encoding the entire CFA/I operon (CfaABCE) has been described previously[29]. Mutations or deletions were introduced into the pETDuet-1-CfaA(Strep) and pMAM2 using the site-directed mutagenesis kit (Quik-Change II XL, Agilent Technologies) to yield the following mutants: pETDuet-1-mtCfaA (Strep), which contains the triple mutations T112I/L114I/V116I, pMAM2-CfaE/R253E, pMAM2-CfaE/(R311-I320) to A, pMAM2-CfaE/S357Y, pMAM2-CfaE/Q358A, pMAM2-CfaE/T359A, pMAM2-CfaE/L360A, pMAM2-CfaE/Δ358–360, and pMAM2-ΔCfaE (a full-length deletion of CfaE). The primers used in the mutagenesis are shown in S1 Table. The mutations and deletions were confirmed by DNA sequencing.

The pMAM2 plasmids containing CfaE mutations were individually transformed into the *E. coli* host strain BL21-AI (Invitrogen), which placed the CFA/I fimbrial operon under control

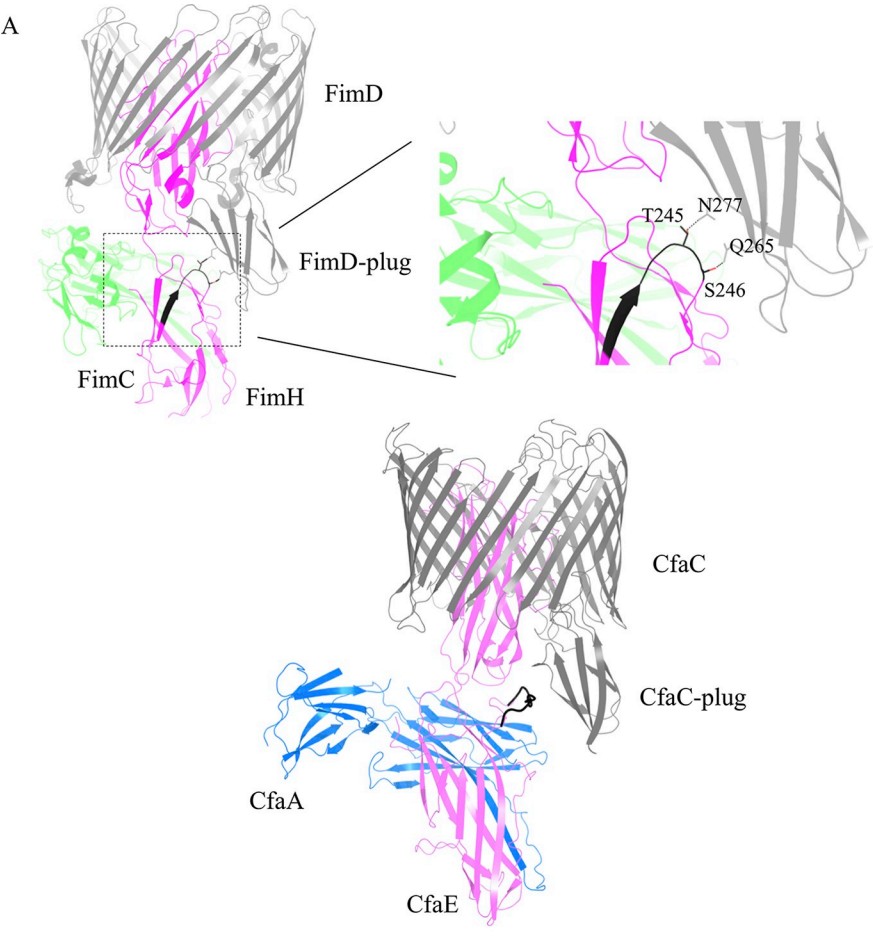

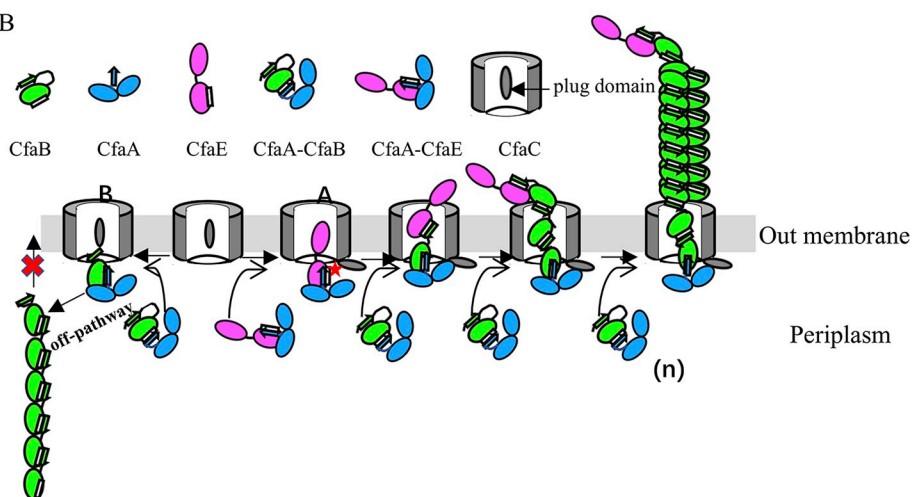

**Fig 6. A model of chaperone-adhesin/usher controlled CFA/I fimbriae assembly.** (A) Top panel: interaction of T245 and S246 in the flexible loop of FimH with N277 and Q265 of the plug domain of FimD revealed in the crystal structure of the quaternary complex of FimD, FimC and FimH (PDB:3RFZ)[38]. FimD, FimC and FimH are in light gray, light green and bright red, respectively. Bottom panel: model of the CfaA (blue)-CfaE(magenta)-CfaC (grey) quaternary complex based on PDB 3RFZ. The functional loops (residues R311-I320 in CfaE, residues L239-S246 in

FimH) are in black. (B) Proposed scheme of the synergistic chaperone-adhesin/usher catalyzed CFA/I fimbria assembly. Two pathways are depicted. Assembly pathway A is a productive pathway, representing a faster, successful CFA/I fimbria biosynthesis. Because CfaA-CfaE has specific interactions with CfaC, the CfaE$_{R311\text{-}I320}$ loop could make contact with the CfaC domain (indicated by red star), allowing the CfaB and CfaE subunits to cross the usher pore. The pathway B represents "off-pathway" assembly in the absence of CfaA-CfaE, in which release of the usher plug domain cannot be accomplished. Therefore, CfaB subunits cannot form a spiral architecture and be transported to cell surface.

of the arabinose-inducible T7 promoter. BL21-AI (pMAM2) and its derivative strains with CfaE mutations were grown in Luria Bertani (LB) medium containing 50 μg mL-1 of kanamycin at 37˚C. When the culture density reached OD$_{600}$ of 1.0, CFA/I fimbrial expression was induced with addition of arabinose to a final concentration of 0.2%. After induction at 37˚C for 90 minutes, cells were collected by centrifugation at 4˚C. Meanwhile, non-induced pMAM2 culture was used as a negative control.

## Protein expression and purification

To express Strep-tagged CfaA in complex with hexahistidine-tagged CfaE or CfaB, the strep-tagged pETDuet-1-CfaAT112I/L114I/V116I (pETDuet-1-mtCfaA) and pCDFDuet-1-(His)6-CfaE or pCDFDuet-1-(His)6-CfaB were co-transformed into *E.coli* BL21 (DE3) cells. The bacterial culture was grown in LB in the presence of 100 μg mL-1 ampicillin and 50 μg mL-1 streptomycin (ZhiChu, Shanghai) shaking at 37˚C until the OD$_{600}$ reached 0.8. The culture was cooled to 16˚C then induced with 0.1 mM isopropyl β-D-1-thiogalactopyranoside (IPTG) for 16 hours. Following induction, the bacteria were collected and resuspended in lysis buffer consisting of 50 mM Tris–HCl, pH 7.5, 150 mM NaCl, 5% glycerol and 1 mM phenylmethane-sulfonyl fluoride (Sigma-Aldrich) and lysed by sonication. The lysate was cleared by centrifugation at 15,000 xg for 45 minutes and then the supernatant was loaded onto a column containing Ni-NTA resin (Qiagen, Germany) for CfaE/CfaB (His$_6$) or Strep-Tactin superflow plus resin (Qiagen, Germany) for mtCfaA (Strep) pre-equilibrated with a binding buffer (20 mM Tris-HCl, pH 7.5 and 100 mM NaCl). Columns were washed with the binding buffer supplemented with 30 mM imidazole in 10 column volumes, and protein sample on the Ni-NTA resin was eluted with the binding buffer with 300 mM imidazole. Protein samples on the Strep-Tactin column were eluted with binding buffer supplemented with 2.5 mM D-desthio-biotin. Affinity purified samples were subjected to size exclusion chromatography (SEC) using an ENrich SEC 650 10 x 300 column (Bio-Rad Laboratories, Inc.). Fractions containing mtCfaA-CfaE/CfaB were pooled and concentrated to a concentration of approximately 8–10 mg mL-1 using a Centricon filter (10 kDa cutoff, Millipore, Billerica) and stored at -80˚C.

To express hexahistidine-tagged CfaC, plasmid pET24-CfaC (His)$_6$ was transformed into *E. coli* BL21strain (DE3). The protein was expressed in the same manner as mentioned above. The lysate was cleared by centrifugation at 6,000 xg for 25 minutes to remove the cell debris. Then the membrane fractions were collected by ultracentrifugation at 100,000xg for 20min. The inner membranes were solubilized in 25 mM Tris–HCl, pH 7.5, 300 mM NaCl containing 0.5% sodium lauroyl sarcosinate for 30 minutes at 4˚C and removed by centrifugation at 100,000 xg for 20 minutes. The precipitate containing outer membranes was resuspended in 25 mM Tris–HCl, pH 7.5, 300 mM NaCl, 5% glycerol and stored at 4˚C prior to further purification. Next, crude CfaC was dissolved in 1% (w/v) n-dodecyl-β-D-maltoside (DDM, Ana-trace) and 0.1% cholesteryl hemisuccinate (CHS, Anatrace). Insoluble materials were removed by ultracentrifugation at 18,000 xg for 60 minutes. The supernatant was mixed with Ni-NTA resin (Qiagen, Germany) equilibrated with a buffer containing 25 mM Tris–HCl, pH 7.5, 300 mM NaCl, 20 mM imidazole, and 0.02% DDM, and gently stirred at 4˚C for 60 minutes. The affinity columns were washed with 10 column volumes of the binding buffer supplemented

with 30 mM imidazole, and protein sample was eluted with the binding buffer supplemented with 300 mM imidazole, and then the eluent was concentrated and further purified by SEC (Superdex 200, GE Healthcare) in a buffer containing 25 mM Tris–HCl, pH7.5, 300 mM NaCl 0.02% DDM. The fractions were determined by SDS-PAGE.

## Crystallization and data collection

Protein crystallization was carried out by the vapor-diffusion method at 20˚C, mixing 1 μL protein with 1 μL well solution. The purified protein was subjected to a high-throughput crystallization screening using a Mosquito crystallization robot (TTP Labtech, UK) and commercially available 96-well kits: Xtal Quest (BioXtal), Wizard (Rigaku), Crystal Screen (Hampton Research) and Index HT (Hampton Research).

Crystals appeared after several days. mtCfaA -CfaE was crystallized using a well solution containing 2% PEG8000, 0.1 M imidazole malate, 1.04 M Lithium Sulfate and 0.001 M GSSG-GSH. All crystals were flash-cooled in liquid nitrogen in the presence of 20–30% glycerol. Diffraction data sets were collected at the SER-CAT ID22 beamline at the Advanced Photon Source (APS), Argonne National Laboratory (ANL) with a MAR300 CCD detector.

## Structure determination and modelling

Diffraction images were indexed and diffraction spots were integrated and scaled using the HKL2000 software package[62]. Structures were solved by molecular replacement with the program suite PHENIX[63]. Subsequent cycles of manual model rebuilding with COOT[64] and refinement with PHENIX improved the quality of the structural models, which were validated using Molprobity[65] before being deposited to the Protein Data Bank.

The model of the CfaC translocation domain was generated by I-TASSER[66]. Template-based complex building was carried out by overlaying the CfaC translocation domain model and CfaE adhesive domain on the corresponding homologous FimH in the FimD-FimC-FimH complex (PDB code 3RFZ).

## Extraction and evaluation of CFA/I fimbria expression

Each gram of cell pastes was suspended in 1 mL phosphate buffered saline (PBS), pH 7.4. After incubation at 65˚C for 20 minutes, cells were removed by centrifugation at 6,000 xg for 30 minutes. The supernatants were collected to evaluate the level of heat-extracted fimbriae from the cell surface by indirect ELISA. Briefly, each well of a 96-well Maxisorp plate (Nunc, Denmark) was coated with 100 μL of samples in triplicate and incubated in 37˚C for 1 hour. Then the plates were washed three times with PBS. After blocking with 200 μL of PBS with 5% fetal bovine serum in 37˚C for 1 hour, wells were washed three times with PBST (PBS containing 0.05% Tween-20). Each well was incubated with 100 μL of primary mouse polyclonal antiserum (1:10,000 dilution) against CFA/I for 1 hour followed by washing five times with PBST. After incubation for 1 hour with 100 μL of a goat anti-mouse horseradish peroxidase (HRP)-conjugated secondary antibody (Jackson ImmounoResearch), the wells were washed again three times with PBST. Finally, the OPD substrate (1 mg mL-1 o-phenylenediamine (Sigma) in sodium citrate buffer (Sigma), pH 4.5 containing 0.4 μL mL-1 of hydrogen peroxide) was added to each well. After incubation for 10 minutes at room temperature, absorbance at 450 nm was recorded for each well using synergy HTX plate reader (Bio-Tek Instruments).

## Acidic Native-PAGE

Acidic Native PAGE (AN-PAGE) was performed according to a previous reported method [67] with modifications. A 10% stacking gel was prepared in 1.5 M acetate-KOH pH 4.3, 50% glycerol, 30% acrylamide, 0.8% methylene bis acrylamide containing TEMED and ammonium persulfate. The 10% resolving gel was prepared in 0.25 M acetate-KOH, pH 6.8. The other components were prepared in the same way as the stacking gel without glycerol. The electrode buffer was 0.14 M acetic acid containing 0.35 M β-alanine, pH 2.9. The loading buffer contained 50% glycerol, 0.25 M acetate-KOH, pH 6.8 and methyl green. The concentrations of purified mtCfaA-CfaE, mtCfaA-CfaB and CfaC were 0.25 μM, 5 μM and 0.25 μM, respectively. Samples were individually incubated for 0, 16, 24, or 36 hours at 25°C. Electrophoresis was performed at 4°C at 120V for 10 hours. The gels were stained with Coomassie blue (0.4% dye made in 50% methanol and 10% acetic acid). Destaining was carried out in a 15% methanol and 10% acetic acid solution.

## Transmission electron microscopy (TEM)

Using TEM to observe the reconstruction of fimbriae in vitro, the concentrations of purified mtCfaA-CfaE, mtCfaA-CfaB and CfaC are 0.25 μM, 5 μM and 0.25 μM, respectively, each sample was incubated at 25°C for 24 hours in a buffer containing 10 mM Tris-HCl, pH 7.5 and 20 mM NaCl, negatively stained with 2% (w/v) uranyl acetate for 1 min and imaged using FEI Tecnai G2 F20 S-TWIN microscope. The micrographs were recorded at an accelerating voltage of 200 kV and a magnification of 20,000. To illustrate the effect of different mutations of CfaE on CFA/I fimbriation by TEM, sample cell pastes were resuspended in PBS, pH 7.4 to a density of 107–108 cells mL-1. Formvar/carbon coated 300 mesh copper grids were prepared by adsorbing 2–5 μL bacterial suspensions for 5 minutes. The grids were then negatively stained with 2% uranyl acetate for 1–2 minutes and imaged using a FEI Tecnai G2 F20 S-TWIN transmission electron microscope. Micrographs were recorded at an accelerating voltage of 200 kV.

## Determining the stability of the complex

Fluorescence emission spectra of purified complex were recorded using an Agilent Cary Eclipse spectrophotometer (Varian, Australia) at 25°C to acquire the equilibrium tryptophan fluorescence spectrum. All experiments were performed in PBS buffer. Spectra were recorded on 1 cm path length cuvettes with an excitation and emission slit width of 2.5 nm. Samples were excited at 295 nm and emission spectra were collected from 300 to 450 nm. Recording for changes in aromatic residue fluorescence at 320 nm began prior to mixing and continued for every 20 ms after mixing (dead time 1s). Different concentrations of GdnCl were used to confirm the stability of the complex. The final concentration of mtCfaA-CfaB or mtCfaA-CfaE was 3.5 μM, the GdnCl concentration was adjusted from 0 M to 6 M and experiments were performed at 0.1 M intervals. Midpoints of dissociation and denaturation phases were obtained by fitting curves to a two-state unimolecular model[68]. The data sets were fitted using GraphPad software Prism 7 (https://www.graphpad.com/).

## MRHA for BL21-AI (pMAM2) and its derivatives with CfaE mutations

Cell pastes of each sample were resuspended in PBS with 0.5% D-mannose (PBSM) to a final $OD_{600}$ of 40. A two-fold dilution series was performed using PBSM as the diluent, and the dilution series was assayed for mannose-resistant hemagglutination (MRHA). In a 12-well ceramic tile plate, to each well was added 25 μL of the bacterial suspension and 50 μL of a 1.5% bovine

erythrocyte suspension, and the plates were incubated on ice with rocking for 20 minutes. Positive MRHA was determined visually by observation of any degree of erythrocyte clumping. The highest bacterial dilution yielding a positive MRHA result was recorded as the MRHA titer. Each experiment was performed in duplicate. Positive and negative control bacteria included BL21-AI (pMAM2) and *E. coli* DH5α, respectively, as well as BL21-AI (pMAM2) without induction.

## Accession numbers

Atomic coordinates of the refined structures have been deposited in the Protein Data Bank (www.pdb.org) with the PDB code 6K73.

## Supporting information

**S1 Fig. SEC profile (Superdex 75, GEHealthcare) and SDS-PAGE of native CfaA-CfaE complex after Ni-NTA purification.** The fraction number is indicated at the top of SDS-PAGE. The results show that the native CfaA-CfaE is not stable and tends to dissociate in solution.
(TIF)

**S2 Fig. The SDS-PAGE of purified CfaC.**
(TIF)

**S3 Fig. The mass spectrometry results verified that the protein ladder bands on AN-PAGE were CfaA-CfaB.**
(TIF)

**S1 Table. Primers used to introduce mutations in pMAM2 in this work.**
(DOCX)

## Acknowledgments

The authors thank the beam line staff of the SER-CAT at APS, ANL for assistance in data collection, George Leiman for editorial assistance, Edward Asafo-Adjei in Division of Pathology at Walter Reed Army Institute of Research for the negative stain electron microscopy of bacteria harboring CfaE mutations, and Guiping Yuan of the Analytical & Testing Center of Sichuan University for transmission electron microscopy for TEM imaging. We thank Public Health and Preventive Medicine Provincial Experiment Teaching Center at Sichuan University and Food Safety Monitoring and Risk Assessment Key Laboratory of Sichuan Province. We thank Professor Yongxing He of Lanzhou University for mass spectrometry identification and analysis. This work was prepared as part of the official duty of S.J.S. and M.G.P. The opinions expressed in this paper are those of the authors and do not reflect the official policy of the Department of the Navy, Department of Defense, or the U.S. Government.

## Author Contributions

**Conceptualization:** Di Xia, Rui Bao.

**Data curation:** Li-hui He.

**Formal analysis:** Hao Wang, Yang Liu.

**Funding acquisition:** Michael G. Prouty, Di Xia, Rui Bao.

**Investigation:** Chang-cheng Li, Yi-bo Zhu, Ying-jie Song.

**Methodology:** Li-hui He, Yang Liu.

**Project administration:** Mei Kang, Ai-ping Tong.

**Resources:** Stephen J. Savarino, Di Xia, Rui Bao.

**Software:** Li-hui He, Tao Li.

**Supervision:** Di Xia, Rui Bao.

**Validation:** Di Xia.

**Visualization:** Li-hui He, Rui Bao.

**Writing – original draft:** Li-hui He, Rui Bao.

**Writing – review & editing:** Li-hui He, Di Xia, Rui Bao.

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
