## [Decision Letter · Decision Letter 0]

20 Jan 2020

Dear Professor bao,

Thank you very much for submitting your manuscript "Chaperone-tip Adhesin Complex Is Vital for Synergistic Activation of CFA/I fimbriae biogenesis" for consideration at PLOS Pathogens. As with all papers reviewed by the journal, your manuscript was reviewed by members of the editorial board and by several independent reviewers. In light of the reviews (below this email), we would like to invite the resubmission of a significantly-revised version that takes into account the reviewers' comments.

Dear Dr Bao,

Thank you for submitting your manuscript "Chaperone-tip Adhesin Complex Is Vital for Synergistic Activation of CFA/I fimbriae biogenesis" to PLoS Pathogens. Your paper has been evaluated by three expert reviewers, whose reports are attached below. As you will see, the reports from the reviewers are quite mixed, and one of the reviewers regards the work not suitable for PLoS Pathogens. I have therefore read your study carefully and inspected the related literature. In view of this assessment and of the positive comments of two reviewers, I decided to offer you the possibility to submit a revised version of your manuscript if you are able to address the reviewers' comments.

Reviewers #1 and #3 have raised multiple issues with the data presentation and interpretation and I recommend that you address their comments in full. Cartoons showing protein structures should be improved for visibility and clear annotations, as indicated by Reviewer #1. Points 4-6 raised by Reviewer #3 require that you undertake additional experiments to satisfy their concerns on the data quantification and reproducibility. In this regard, please refer to the data availability policy of PLoS Pathogens: https://journals.plos.org/plospathogens/s/data-availability. Raw data used to produce graphs and uncropped gel images should be made available as supplementary information or via DOI in public repositories.

Please address Reviewers' #1 and #3 concerns on the quality of EM images, notably in Figure 1C, which at the present level of resolution do not support the claims on different pilus states - coiled versus extended. Please provide higher quality images and insets with higher magnification to clearly demonstrate the dimensions/diameters of different pili and show any differences between them. Pilus length measurements similar to those performed in Nishiyama et al 2008 would provide insight into pilus polymerization defects. Please indicate clearly the correspondence between samples in Figure 1B and 1C. It is unclear what the designation "uncatalyzed" refers to in your study.

In the EM analysis of bacteria with mutations in CFA/I components, please indicate clearly which appendages correspond to CFA/I fimbriae and which ones are other pili/flagella. Please revise the model in Figure 6 following Reviewers #1 and #3 comments to summarize data described in the present study. Kinetic constants are not part of this research. Also, in the "off" pathway model, what is the reason that polymerization of CfaB fails to produce a coiled rod polymer in vitro? Are not all determinants of its molecular spring behaviour present in the major subunit (e.g. Spaudling et al eLife 2018 )? Finally, may I request that you cite the appropriate original literature, notably reports obtained from related CU pili studies that set the paradigms for assembly mechanism, role of adhesin in pilus assembly initiation, usher dynamics, pilus rod spring-like properties etc.

If you can respond to all of the referees’ points - by making the requested changes or by providing a compelling argument why a change cannot or should not be made - then I encourage you to submit a revised manuscript. In light of the further experimental work requested by reviewer #3, I am handling this as a Major Revision with the understanding that the revised manuscript will be subject to a full review by these same (and possibly new) referees. Please note that acceptance of your revised manuscript is not guaranteed. In general, revised manuscripts should be returned within three months. If you anticipate that significantly more time will be needed, please let me know.

We cannot make any decision about publication until we have seen the revised manuscript and your response to the reviewers' comments. Your revised manuscript is also likely to be sent to reviewers for further evaluation.

Sincerely,

Olivera Francetic, PhD

Guest Editor

PLOS Pathogens

Guy Tran Van Nhieu

Section Editor

PLOS Pathogens

Kasturi Haldar

Editor-in-Chief

PLOS Pathogens

orcid.org/0000-0001-5065-158X

Michael Malim

Editor-in-Chief

PLOS Pathogens

orcid.org/0000-0002-7699-2064

Dear Dr Bao,

Thank you for submitting your manuscript "Chaperone-tip Adhesin Complex Is Vital for Synergistic Activation of CFA/I fimbriae biogenesis" to PLoS Pathogens. Your paper has been evaluated by three expert reviewers, whose reports are attached below. As you will see, the reports from the reviewers are quite mixed, and one of the reviewers regards the work not suitable for PLoS Pathogens. I have therefore read your study carefully and inspected the related literature. In view of this assessment and of the positive comments of two reviewers, I decided to offer you the possibility to submit a revised version of your manuscript if you are able to address the reviewers' comments.

Reviewers #1 and #3 have raised multiple issues with the data presentation and interpretation and I recommend that you address their comments in full. Cartoons showing protein structures should be improved for visibility and clear annotations, as indicated by Reviewer #1. Points 4-6 raised by Reviewer #3 require that you undertake additional experiments to satisfy their concerns on the data quantification and reproducibility. In this regard, please refer to the data availability policy of PLoS Pathogens: https://journals.plos.org/plospathogens/s/data-availability. Raw data used to produce graphs and uncropped gel images should be made available as supplementary information or via DOI in public repositories.

Please address Reviewers' #1 and #3 concerns on the quality of EM images, notably in Figure 1C, which at the present level of resolution do not support the claims on different pilus states - coiled versus extended. Please provide higher quality images and insets with higher magnification to clearly demonstrate the dimensions/diameters of different pili and show any differences between them. Pilus length measurements similar to those performed in Nishiyama et al 2008 would provide insight into pilus polymerization defects. Please indicate clearly the correspondence between samples in Figure 1B and 1C. It is unclear what the designation "uncatalyzed" refers to in your study.

In the EM analysis of bacteria with mutations in CFA/I components, please indicate clearly which appendages correspond to CFA/I fimbriae and which ones are other pili/flagella. Please revise the model in Figure 6 following Reviewers #1 and #3 comments to summarize data described in the present study. Kinetic constants are not part of this research. Also, in the "off" pathway model, what is the reason that polymerization of CfaB fails to produce a coiled rod polymer in vitro? Are not all determinants of its molecular spring behaviour present in the major subunit (e.g. Spaudling et al eLife 2018 )? Finally, may I request that you cite the appropriate original literature, notably reports obtained from related CU pili studies that set the paradigms for assembly mechanism, role of adhesin in pilus assembly initiation, usher dynamics, pilus rod spring-like properties etc.

If you can respond to all of the referees’ points - by making the requested changes or by providing a compelling argument why a change cannot or should not be made - then I encourage you to submit a revised manuscript. In light of the further experimental work requested by reviewer #3, I am handling this as a Major Revision with the understanding that the revised manuscript will be subject to a full review by these same (and possibly new) referees. Please note that acceptance of your revised manuscript is not guaranteed. In general, revised manuscripts should be returned within three months. If you anticipate that significantly more time will be needed, please let me know.

Reviewer's Responses to Questions

**Part I - Summary**

Reviewer #1: This manuscript is an important contribution to the field of bacterial adhesion and in the longer term, use of fimbriae in vaccine development. The authors provide new data, rigorously obtained, that advance our understanding of ETEC adhesion, which is essential for advancing efforts to prevent diarrheal diseases.

However, the authors discuss kinetics of assembly as if they are presenting kinetic data, which do not appear to be part of this research. The distinction between data-supported statements and speculation on their part needs to be better delineated.

Reviewer #2: The paper " Chaperone-tip Adhesin Complex Is Vital for Synergistic Activation of CFA/I fimbriae biogenesis" by Li-hui He and colleagues describes an in vitro assay to form CFA/I fimbria assembly and provide insight to the process that drives fimbria formation. The scope of this work is indeed very important for the fimbriae community, and I find the conclusions exciting and novel. Overall, the work is described clearly in the manuscript, and the results are strongly supported by data, and the authors' methods are sound and strong.

Reviewer #3: The manuscript by He et al. examines the assembly mechanism of CFA/I fimbriae expressed by enterotoxigenic E. coli. CFA/I fimbriae are major colonization factors and are assembled by the chaperone-usher (CU) pathway. The authors extend previous structure-function analyses of these fimbriae by mutating the CfaA chaperone to allow for purification of stable chaperone-subunit complexes with CfaB (major subunit protein) and CfaE (tip-localized adhesin subunit). Using this system, the authors establish an in vitro assay to monitor fimbrial polymerization and they generate a crystal structure of the CfaB-CfaE complex. Overall, while the authors present important new findings for the CFA/I fimbriae, the manuscript is not well organized or presented, the data seem preliminary and insufficient to support the conclusions, and the results need to be placed in better context of what is known from structural/functional analyses of other CU systems. Additional comments on the manuscript are listed below.

**Part II – Major Issues: Key Experiments Required for Acceptance**

Reviewer #1: While no key experiments are required for acceptance, it is essential that the authors clarify their statements on the kinetics of fimbria assembly - what data support their analysis, and what is speculation?

Reviewer #2: I find the experiments performed in this study strong!

Reviewer #3: 1. The manuscript text and figures need significant improvements for organization, presentation and clarity. Improvements are also needed for better citation of references and to correct errors throughout.

2. Many of the conclusions are not sufficiently supported by the data and are excessively speculative in nature, including the model presented in Fig. 6. The authors do not sufficiently incorporate or acknowledge the extensive prior structural and functional work done on CU pathways.

3. Better description is needed for what was previously known regarding the triple CfaA mutant chaperone used to stabilize the CfaA-B and CfaA-E complexes in this study. Has this specific mutant chaperone been demonstrated to be functional for assembly of CFA/I fimbriae in bacteria? More information is also needed regarding the CfaA-B crystal structure used in this study for comparison with the CfaA-E structure. Is the crystal structure of the CfaA-B complex from reference 10 (Bao et al, 2016)? That structure uses CfaA with a single mutation (T112I) rather than a triple mutation (correct?). If the chaperones are different, then this complicates the direct comparisons made in the current study.

4. More data is needed to support the in vitro fimbrial polymerization assay. The gel and EM images are of poor quality and the differences in fimbrial fibers stated in the text (length of fibers and extended vs. helical forms) are not apparent in the EM images. Also, more time points are needed to look between the 0 and 16 h reaction times and quantitative analysis is needed of the reaction to support the authors’ conclusions.

5. The denaturation/fluorescence data used to probe thermodynamic stability of the CfaA-B and CfaA-E complexes are difficult to interpret and not convincing (and the data should be moved to the main text rather than in the supplemental information). These data are insufficient to support the authors’ conclusions regarding relative stabilities of the complexes.

6. The analyses of the CfaE mutants – agglutination, ELISA, EM – also need improvements, including sufficient biological replicates to allow for statistical analyses and better EM images.

**Part III – Minor Issues: Editorial and Data Presentation Modifications**

Reviewer #1: More minor points include figures that are difficult to interpret - most drawings of the models have lines that are too thin to be easily followed, and the point being made in the text is sometimes not clearly shown in the figure. Specific examples are shown below, but the authors should also look at all the figures and try to make them easier to understand without significant zooming.

The authors state “CFA/I fimbriae exhibit a spring-like property, also

71 known as catch-bond, that allows persistent attachment to the intestinal

72 epithelia even under the conditions of shear stress[4].”

Catch-bonds are not ‘spring-like’, so this sentence appears to be incorrect.

The authors state “the chaperone captures an unfolded pilus subunit in the

79 periplasm via the donor-strand complementation (DSC) mechanism”

Is the subunit really unfolded before attaching to the chaperone? Is this known?

L150-152 is awkward. It would be easier to read if it said ‘only in the presence of both’ rather than simply ‘both’

Table 1 ‘Unit Cell’ Numbers in parentheses are not explained.

Fig 2C - I cannot determine what the authors are trying to show. Is this a perspective view rather than orthographic?

The very thin lines of this figure are difficult to see

L271-272. Unclear where the C-term of the chaperone is. Having just a closeup in fig 3 does not help orient the reader

Fig 4A does not show about which axis the domains are rotated. The lines are too thin to be visible, and the yellow has very little contrast.

L337 please clarify ‘preserves its folding energy for assembly initiation’ .

This appears to be the authors’ speculation, not a result.

L 339-343 needs a figure to visualize this change.

Fig 4C is not understandable. First, where are the beta strands that are discussed in the text? How is the looped ‘wedged between’? Second, is ‘gray’ the tan or the blue? Whichever it is, what is the other color? The atoms shown in blue/red are confusing, because then it is difficult to tell whether they are from the magenta or tan subunit.

The whole figure could be re-thought. Perhaps have them displayed separately and then together?

Table 3 - terminal is mis-spelled

L404 ‘and’ should be ‘or’

L405 it is unclear what the authors mean by saying these residues ‘may play a regulatory role in interacting with the chaperone’. In addition, this statement belongs in the discussion, not results.

L418 should be Table 3

Fig 5 please state the scale bar size in the legend — it is not legible even at 3x zoom

Are the thicker filaments bundled pili, or do these bacteria have flagella?!

Fig 1C and Fig 5 - The electron micrographs shown are acceptable with respect to showing the data they are said to show, but they are of poor quality and detract slightly from the manuscript.

L459-463 why would it be paradoxical that the less stable is also less stable as a heterodimer?

L463-467 I don’t see the logic that leads the authors to conclude that a subunit with more ‘strain’ would be less stable.

L 477 It is not supported that assembly beginning with CfaE is due to the reasons stated, that ‘correct initiation may need more input energy and that elongation is rapid after initiation. Indeed, the fact that there is no assembly without CfaE means that ‘correct initiation’ is always the case, and there are no data in the manuscript regarding the kinetics of assembly.

L 495-499 clarifies statements made earlier in the manuscript, so perhaps should move there (L 377; see above).

L504 what does ‘insufficient’ mean in this context?

L508-509 ‘fits the energy profile of the CfaA-CfaE complex’. What energy profile? Are there data to support this statement?

Fig 6 - red star is not visible except after significant zoom.

L783-784 please state what structure (which proteins) was deposited as pdb 6K73. If it is CfaA/CfaE as stated in the abstract, then what is shown in Fig 6B? Please include the pdbs of all the structures shown in Fig 2B-2D

Fig 3 it is confusing to have the subunits be blue and magenta, and then the atoms be colored as blue and red, irrespective of which subunit they are from.

Fig S2 the x-axis is mislabeled - it should be GdnCl. Also, there should be vertical lines marking the positions of the 2 inflection points discussed in the text.

Reviewer #2: I found the following mistake in the introduction on page 4, row 70-73: “CFA/I fimbriae exhibit a spring-like property, also known as catch-bond, that allows persistent attachment to the intestinal epithelia even under the conditions of shear stress[4].”

A catch-bond does not give rise to spring-like properties! The spring-like properties of CFA/I, that is the ability to unwind and rewind the shaft subunits, was shown in Prof. Bullitt’s work in 2012, A Structural Basis for Sustained Bacterial Adhesion: Biomechanical Properties of CFA/I Pili. Journal of Molecular Biology, 415(5), 918–928. A catch-bond modulaters the bond-receptor interaction since they can dynamically change the bond-strength when exposed to a tensile force. A catch-bond behaves, therefore, more like a finger-trap, the more load that is applied to the bond, the life-time of the bond-receptor complex will increase, up to the threshold break force.

Reviewer #3: (No Response)

PLOS authors have the option to publish the peer review history of their article (what does this mean?). If published, this will include your full peer review and any attached files.

Reviewer #1: Yes: Esther Bullitt

Reviewer #2: No

Reviewer #3: No
---

## [Decision Letter · Decision Letter 1]

15 Jul 2020

Dear Professor bao,

Thank you very much for submitting your manuscript "Chaperone-tip Adhesin Complex Is Vital for Synergistic Activation of CFA/I fimbriae biogenesis" for consideration at PLOS Pathogens. As with all papers reviewed by the journal, your manuscript was reviewed by members of the editorial board and by independent reviewers. The reviewers appreciated the attention to an important topic and agree that the manuscript has been substantially improved, by addressing the comments of the three initial reviewers. However, there are multiple issues related to writing and data presentation that need to be addressed before the paper can be accepted for publication.

One of these issues concerns the figures, which have now been split into multiple panels. I will ask you to kindly follow the guidelines for the authors and to submit one original file for each figure, with each panel labeled to facilitate reference to the panels in the main text. In the case of Figure 1, which has been substantially modified, I suggest that you keep a similar mode of organisation as in the original submission, with the 4 figures showing the same type of in vitro assay data for four different protein samples grouped into one panel: SDS-PAGE data in Panel B, quantifications in Panel C, microcopy in D and measurements in E. Alternatively, you may choose to group the assay data and quantifications into a single panel and the EM images and quantifications in another. The figure legends should be modified accordingly and they should explain all the parts of the figures (this is not the case in the current version). For example, in the bottom panel of Fig 1B-1E, it is unclear why there is an inset in each panel with the quantification on a different scale. Are these different representations of the same data ? Figure legends should provide more details on these graphs. Please refer to the figures by "Fig 1A" instead of "Figure. 1A" etc. throughout the text. Also, please follow the same order of figures as that of the analyses presented in the main text. Thus, I suggest that the order of figures 4 and 5 be inverted, since you first refer to the data presented in Fig 5 in the main text, followed by those shown in Fig 4.

Finally, please make the necessary changes in the text as detailed below in the editor's comments and as suggested by Reviewer #1.

Sincerely,

Olivera Francetic, PhD

Guest Editor

PLOS Pathogens

Guy Tran Van Nhieu

Section Editor

PLOS Pathogens

Kasturi Haldar

Editor-in-Chief

PLOS Pathogens

orcid.org/0000-0001-5065-158X

Michael Malim

Editor-in-Chief

PLOS Pathogens

orcid.org/0000-0002-7699-2064

Minor editor's comments:

Line 63. We demonstrate

Line 64. Replace “play indispensable roles” with “is essential for “

Line 65. “The crystal structure of this complex reveals…”

Line 67. Remove “moreover” – Our findings suggest…

Line 69. …facilitates…

Line 70. …lumen. Collectively, our data demonstrate the critical role…

Line 117. “..tip-localized adhesins are required for initiation of assembly of various fimbriae…(24-29), including the Class 5 fimbriae (30)”

Line 131. “…mutant CfaA variant to stabilize…”

Line 137. Our work shows that CfaE is required for…

Line 157-160. When compared to native CfaA, the mutant variant containing the triple residue substitution T112I/L114I/V116I on the G1 strand (hitherto referred to as mtCfaA) formed significantly stabilized complexes with CfaE (Fig 1A), CfaB, and CfaBntd (the CfaB variant lacking the N-terminal donor strand).

Line 162. , 164 etc. (Fig S2) please correct all references to figures in this manner.

Line 168. (Fig 1B to Fig 1E top two panels) – referring to the figure in this way is confusing. Please show the data as a single figure file, with panels regrouped and labeled. The data from the current panels 1B to 1E should be regrouped.

Line 178 …was able to prevent…

Table 1: Unit cell parameters

Line 213. … more open cleft…

Line 219. …shorter life-span compared to that of the CfaA-CfaB complex in solution.

Line 225. CfaA variant with triple residue substitution in the G1 strand (redundant as it has been introduced above)

Line 241. … allowed us to compare…

Line 260. … likely to engage in CfaA interaction (Fig 3A)

Also, please replace “C-terminal” by “C-terminus” in the figure 3A.

Line 269. Figure 5A data are described in the main text before Figure 4. The order of these figures should be inverted.

Line 381. Remove “the establishment” from the title and replace by "An in vitro system… etc"

Line 386. This mutant variant is able to…

Line 417. … suitable confirmation…

Line 419 …very rapidly…

Line 423. While mutations in these CfaE elements did not impair binding to CfeA, they affected assembly.

Line 425. …and not necessarily …

Line 452 ...do not require…

Line 455 …for the usher…

Line 456. … whether this motif is responsible….

Line 466 … show that the….

Line 467. … domain binds more tightly to the usher pore in the initiation complex compared to the elongation complex…

Line 468. Based on those….

Line 471. ....and facilitates binding of the CfaE adhesin domain to the CfaC lumen

line 623. concentrations ... were 0.25 etc.

line 647. …different mutations...

line 668. .. performed at 0.1M intervals

Line 686. modify the figure and the legend in the new version. Please explain the purpose of the two different scales and insets of the graphs on the bottom panels?

Line 691. ...ladders resulted from aggregation of...

Line 695... representative CfaB polymers...

Line 714. ... (dashed line) ... fitted curves generated by GraphPad...

Line 733. should be: "Superimposed cartoon models of the pilin domain CfaE from...

Line 737. The disordered loop 301-321 in mtCfaA-CfaE is shown as dashed line

Line 743. ... involved in domain interactions

Line 751. Remove the statement about the zoom.

Line 761. Two pathways are depicted.

Reviewer Comments (if any, and for reference):

Reviewer's Responses to Questions

**Part I - Summary**

Reviewer #1: The authors have successfully addressed all comments from the initial review, and the manuscript is acceptable for publication.

**Part II – Major Issues: Key Experiments Required for Acceptance**

Reviewer #1: none.

**Part III – Minor Issues: Editorial and Data Presentation Modifications**

Reviewer #1: L 63 demonstrate

L 64 plays

L 65 reveals

L 69 facilitates

L 140 functions

L 158 Figure

L 171 delete 'Fim'

L 178-179 I think you mean prevent self-assembly into off-pathway trimers -- it is 'self'assembing' (polymerizing) into short fimbria

L346-347 it is not negative-stain EM, but EM of negatively stained fimbria

L468 Based

L471 lumen

L 751 It is not clear what is meant by 'the zoom is 10,000x'. Was the image as sent to the journal zoomed 10kx? Since how it will be viewed or printed is not known, this is not useful information. If the authors would like to include something more than the scale bar, they can include the magnification at which the images were recorded.

PLOS authors have the option to publish the peer review history of their article (what does this mean?). If published, this will include your full peer review and any attached files.

Reviewer #1: **Yes: **Esther Bullitt
---

## [Editor Report · Decision Letter 2]

24 Jul 2020

Dear Professor bao,

Thank you very much for submitting the revised version of your manuscript "Chaperone-tip Adhesin Complex Is Vital for Synergistic Activation of CFA/I fimbriae biogenesis" for consideration at PLOS Pathogens.

Before we ca proceed with the final acceptance of your paper, there are still a few issues remaining. At this stage, it is important to follow the journal guidelines for manuscript formatting. In particular, I will kindly ask you to submit also all the other figures (Fig. 2 to Fig. 5) as single .tiff files and not as individual panels, just as you had done already for Figure 1. Also please submit all the Tables as separate files, and please follow the journal guidelines in terms of format, font, resolution etc.

PLOS Pathogens does not provide a copy editor service, therefore it is the authors' and editor’s responsibility to ensure that the final version is published free or errors, to avoid subsequent corrections and errata.

In this regard, the following corrections remain to be made:

Line 70. « lumen » instead of « luman »

Line 117. “Localized” instead of “locatized”

Line 177. "…and to extend CfaB polymer…”

Line 196. Table 1. “Unit cell” instead of “unite cell…” and average B factor (Å²) instead of (Å2)

Line 223. Change to: “…variant with triple residue substitution in the G1…”

Line 257. Please replace “C-terminus” with “C-terminal” here in the text, going back to the previous version. The label “C-terminal” should have been replaced by “C-terminus” in the Figure 3A itself, not in the text.

Please ensure that the figure 3, panel is modified accordingly this time.

Line 344. "The ΔCfaE mutant had no…"

Line 345. “…had a negative MRHA phenotype”

Line 357. “.. production was reduced to the uninduced levels…”

Line 360. “Mutations of the three C-terminal residues…”

Line 364-365. “Indeed, a single R253E substitution in CfaE led to the loss of CFA/I fimbriation by negative-stain EM and to negative MRHA…”

Line 373-374. “… which corroborated the results of bact-ELISA…”

Line 380. Remove the italics for “system”

Line 476. “development” instead of “devolvement”

Please verify the completeness of these modifications so as to avoid new rounds of revision. Thank you for your understanding.

Sincerely,

Olivera Francetic, PhD

Guest Editor

PLOS Pathogens

Guy Tran Van Nhieu

Section Editor

PLOS Pathogens

Kasturi Haldar

Editor-in-Chief

PLOS Pathogens

orcid.org/0000-0001-5065-158X

Michael Malim

Editor-in-Chief

PLOS Pathogens

orcid.org/0000-0002-7699-2064
---

## [Editor Report · Decision Letter 3]

30 Jul 2020

Dear Professor bao,

We are pleased to inform you that your manuscript 'Chaperone-tip Adhesin Complex Is Vital for Synergistic Activation of CFA/I fimbriae biogenesis' has been provisionally accepted for publication in PLOS Pathogens.

Best regards,

Olivera Francetic, PhD

Guest Editor

PLOS Pathogens

Guy Tran Van Nhieu

Section Editor

PLOS Pathogens

Kasturi Haldar

Editor-in-Chief

PLOS Pathogens

orcid.org/0000-0001-5065-158X

Michael Malim

Editor-in-Chief

PLOS Pathogens

orcid.org/0000-0002-7699-2064
---

## [Editor Report · Acceptance letter]

4 Sep 2020

Dear Professor Bao,

We are delighted to inform you that your manuscript, "Chaperone-tip Adhesin Complex Is Vital for Synergistic Activation of CFA/I fimbriae biogenesis," has been formally accepted for publication in PLOS Pathogens.

Best regards,

Kasturi Haldar

Editor-in-Chief

PLOS Pathogens

orcid.org/0000-0001-5065-158X

Michael Malim

Editor-in-Chief

PLOS Pathogens

orcid.org/0000-0002-7699-2064